# Rethinking Parameter Sharing as Graph Coloring for Structured Compression

**Boyang Zhang** [1 2 3]  **Daning Cheng\*** [1]  **Yunquan Zhang** [1]  **Fangming Liu** [2]

zhangby01@pcl.ac.cn, {chengdaning, zyq}@ict.ac.cn, fangminghk@gmail.com,

## Abstract

Parameter sharing is a key model compression technique, yet existing methods overlook the geometric properties of the loss landscape, often causing severe accuracy degradation under high compression ratios. Inspired by second-order optimization, we propose Curvature-aware Graph Coloring (CGC), a cross-layer parameter sharing framework that treats each network layer as a graph node, with each node assigned to a shared low-rank basis. CGC leverages Hessian eigenspace information to group layers with similar curvature profiles, aligning the perturbations introduced by parameter sharing with the low-curvature (flat) directions of the loss ellipsoid. This effectively mitigates performance loss while enabling flexible, global cross-layer sharing. Experiments on LLaMA-7B and Swin Transformer show that CGC achieves 28%–50% parameter compression with Top-1 accuracy loss no more than 0.01% on Swin—or even accuracy gains on LLaMA—while delivering over 60% higher inference throughput, significantly outperforming SVD-based and heuristic-based methods. This work demonstrates that curvature-guided, geometry-aware sharing is essential for efficient, stable, and high-ratio model compression.

## 1. Introduction

Parameter sharing is pivotal in model compression (Zhang et al., 2026). But current parameter-sharing approaches still face a fundamental performance challenge: the accuracy loss after compression is difficult to control, especially under high compression ratios where performance often degrades sharply. The root cause is that existing methods

[1]Institute of Computing Technology, Chinese Academy of Sciences, Beijing, China [2]Pengcheng Laboratory, Shenzhen, China [3]University of Chinese Academy of Sciences, Beijing, China. Correspondence to: Daning Cheng <chengdaning@ict.ac.cn>.

*Proceedings of the 43$^{rd}$ International Conference on Machine Learning*, Seoul, South Korea. PMLR 306, 2026. Copyright 2026 by the author(s).

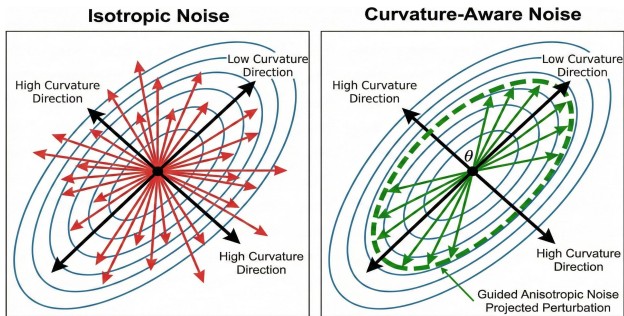

*Figure 1.* Geometric interpretation of sharing noise. Existing methods fail to control the perturbation direction, resulting in isotropic noise (red) that inevitably intersects high-curvature axes. In contrast, our method guides perturbations along low-curvature directions (green), confining them to "flat" regions.

predominantly rely on rigid heuristic rules to design sharing structures—for example, enforcing sharing only between adjacent layers, partitioning weights into fixed-size blocks, or manually grouping layers by index. These strategies ignore the true distribution of parameter redundancy within the network and fail to account for the model's differential sensitivity to perturbations along different directions. Consequently, even if the introduced perturbation is small in Euclidean norm, it can still cause significant performance degradation if aligned with high-sensitivity regions of the loss landscape.

Existing works (Hassibi et al., 1993; Frantar et al., 2023; Zhang et al., 2025) show that the noise induced by parameter sharing can be effectively characterized by second-order information—specifically, the Hessian matrix. When a small perturbation is applied to the parameters, the resulting increase in loss is primarily determined by the projection of that perturbation onto the Hessian's eigenspace. Locally, the loss surface can be approximated as a multidimensional ellipsoid: its minor axes correspond to high-curvature directions (large eigenvalues), where the loss is highly sensitive to parameter changes; in contrast, its major axes correspond to low-curvature directions (small eigenvalues), representing flat regions where parameter variations have minimal impact on performance.

Therefore, an ideal compression-induced perturbation should be aligned as closely as possible with the ellipsoid's

major axes. This not only maximizes tolerance to parameter changes but also ensures stable performance. As illustrated in Figure 1, while traditional methods introduce isotropic noise (red arrows) that inevitably intersects high-curvature contours, our approach confines perturbations to the low-curvature valley (green arrows), effectively "sliding" along the flat directions of the loss landscape.

Motivated by this geometric insight, we propose Curvature-aware Graph Coloring (CGC), a novel cross-layer parameter sharing framework. We treat each layer of the model as a node in a graph and formalize the design of the sharing structure as a graph coloring problem: each "color" corresponds to a shared low-rank basis, and the coloring process assigns each layer to the shared basis that induces the smallest high-curvature perturbation under a trust-region constraint. Unlike traditional heuristic approaches, CGC explicitly leverages Hessian eigenspace from each layer as the matching criterion when constructing and selecting shared bases. Specifically, by analyzing the curvature profile of each layer's loss landscape, we identify its dominant low-curvature directions and enforce alignment with these directions during basis assignment. This ensures that the perturbation introduced by parameter sharing predominantly resides near the major axes of the loss ellipsoid.

This design yields distinct advantages. First, perturbations are actively steered toward directions least harmful to performance, significantly mitigating accuracy loss. Second, because the coloring process is no longer constrained by topological proximity, our method naturally supports flexible, arbitrary cross-layer sharing. This allows it to uncover deep redundancies across the entire network and transcend the limitations of adjacent-layer sharing. Furthermore, by formalizing sharing through graph theory, our framework implicitly captures the structural symmetry of the parameter, offering a rigorous mathematical grounding for redundancy reduction. Our key contributions are summarized as follows:

• **Geometric-Algebraic Framework**: We formalize cross-layer parameter sharing as a graph coloring problem. By incorporating automorphism groups to characterize structural symmetry, we provide a theoretical guarantee for capturing deep inter-layer redundancies beyond simple heuristics.

• **Curvature-aware Mechanism**: We propose a novel coloring function $\alpha$ that leverages Hessian analysis. This mechanism actively aligns shared parameters with low-curvature directions, avoiding the exponential complexity of combinatorial search while minimizing accuracy degradation.

• **Performance**: We validate CGC on LLaMA-7B and Swin Transformer. At 28%–50% compression rates, CGC maintains Top-1 accuracy loss within 0.01% (or achieves gains) and boosts inference throughput by over 60%, significantly outperforming SVD-, and heuristic-based sharing baselines.

## 2. Related Work

**SVD-based Weight Decomposition.** As deep models grow increasingly large, their memory and compute overhead at inference time severely limits deployment in edge scenarios, where resources are constrained. Model compression is therefore crucial for enabling practical deployment. Singular value decomposition (SVD) and low-rank approximation are widely adopted techniques for neural network compression (Golub et al., 1987; Vaswani et al., 2017; Lv et al., 2023; Wu et al., 2023; Hsu et al., 2022). There are many works to improve the performance of SVD based methods (Zhang et al., 2024; Wang et al., 2025c; Yuan et al., 2025). Despite these advances, most methods treat each layer independently, overlooking structural redundancies that could be exploited across layers—a limitation addressed by CGC.

**Parameter Sharing.** Parameter sharing (Wang et al., 2025b) reduces model size by reusing weights across layers. Universal Transformer (Dehghani et al., 2018) shares weights entirely across encoder and decoder layers, while Subformer (Reid et al., 2021) divides parameters into attention and feed-forward groups for intra-group sharing. Most methods adopt group-wise strategies, where identical weights are shared within predefined groups. Dynamic Tying (Hay & Wolf, 2024) explores sharing structures during training via reinforcement learning, but its computational cost limits scalability to large models. Training-free methods like FiPS (Üyük et al., 2025) minimize block-level reconstruction error for ViTs, yet restrict sharing to adjacent blocks. Basis Sharing (Wang et al., 2025a) improves this by representing adjacent layers with shared basis and coefficients. However, these methods rely on heuristics and fail to explore comprehensive sharing across multiple layers—a gap addressed by CGC, which systematically identifies shared structures spanning layers to optimize compression.

## 3. Approach

### 3.1. Curvature-aware Graph Coloring

Deep neural networks, upon convergence, exhibit a distinct geometric structure in the loss landscape: locally, the loss surface can be approximated by an ellipsoid, where the major axes correspond to low-curvature (flat) directions and the minor axes to high-curvature (sensitive) directions. This geometric insight reveals a key principle—if parameter perturbations introduced during model compression can be steered toward flat directions, significant parameter reduction can be achieved while preserving model performance.

Motivated by this observation, we propose Curvature-aware Graph Coloring (CGC), shown in Algorithm 1. The pipeline of CGC is shown in Figure 2. CGC treats each network layer as a node in a graph and formulates cross-layer parameter sharing as a graph coloring problem: each "color"

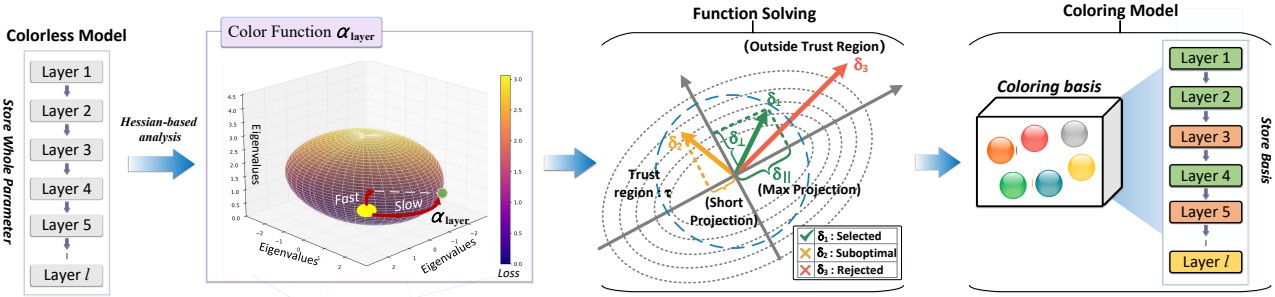

*Figure 2.* Curvature-aware Graph Coloring (CGC) pipeline: Starting from the original parameters (Colorless Model), we analyze the local loss geometry via Hessian analysis to identify high- and low-curvature directions. In the Function Solving stage, the algorithm selects a perturbation $\delta$ that aligns with the flat directions to minimize error, strictly constrained within a trust region $\tau$ (e.g., $\delta_1$ is selected as optimal, while $\delta_3$ is rejected for violating the constraint). Finally, layers assigned to the same basis are grouped in the Coloring Model.

represents a shared low-rank basis, and the coloring process aims to assign layers with similar curvature characteristics to the same color group. Crucially, we characterize the local geometry of each layer using the eigenspace of its Hessian matrix and measure inter-layer similarity based on their low-curvature principal directions. During basis assignment, the algorithm prioritizes alignment along these flat directions, ensuring that sharing-induced perturbations primarily reside in the subspace where the loss changes most slowly. This curvature-guided matching naturally confines compression error along the ellipsoid's major axes, enabling stable and low-loss compression. Moreover, the graph coloring framework supports flexible, global grouping across the entire network—without relying on predefined adjacency or fixed patterns—significantly enhancing redundancy exploitation and compression efficiency.

### 3.2. Parameter Sharing as Graph Coloring

We use the term graph coloring in a coloring-inspired sense rather than as classical vertex coloring. In classical graph coloring, adjacent vertices are constrained to receive different colors. In CGC, each color denotes a shared low-rank basis, and the objective is to assign each layer to the basis that minimizes a curvature-aware perturbation cost.

We formulate cross-layer parameter sharing as a geometry-guided graph coloring problem. Specifically, each layer of the neural network is treated as a node in a graph. An edge is implicitly established between two layers if they exhibit similar local geometric properties in the loss landscape—namely, comparable sensitivity to parameter perturbations—indicating their potential to share a common structural basis. The goal is to assign a "color" to each node, where each color corresponds to one shared basis. This global perspective enables our method to move beyond heuristic constraints such as "adjacent-layer-only" sharing and automatically discover functionally redundant structures that recur across the entire model depth.

Crucially, CGC does not enforce identical weight values across layers, which would severely limit expressivity. We therefore follow a more flexible strategy: sharing the underlying structural properties of weight matrices instead of their exact values. To achieve this, we employ low-rank decomposition. Each shared basis $B_b$ is defined as a pair of factor matrices $B_b = (U_b, V_b)$. A layer $\ell$ that is assigned to this basis constructs its weight matrix $W_\ell$ by combining the shared basis with a small, layer-specific coefficient matrix $S_{\ell,b}$: $W_\ell \approx U_b S_{\ell,b} V_b^T$. Here, $U_b \in \mathbb{R}^{M \times r}$ and $V_b \in \mathbb{R}^{N \times r}$ are the shared factors, while $S_{\ell,b} \in \mathbb{R}^{r \times r}$ preserves layer-specific information. The rank $r$ acts as a trade-off between compression and expressivity. The task thus becomes finding an optimal coloring function. Specifically, we treat this as an assignment problem on a complete graph. The objective is not to satisfy adjacency constraints, but, for a given set of candidate basis $\mathcal{B}$ (generated following (Wang et al., 2025a)), to find an assignment function $\alpha_{\text{layer}}$ that maps each layer to a basis in $\mathcal{B}$ that minimizes performance impact.

By constraining same-color layers to share identical basis factors $(U, V)$, this approach induces structural symmetry, rendering these layers functionally interchangeable. We observe that these symmetric configurations naturally gravitate toward flatter regions of the loss landscape, even without explicit optimization. This indicates that our method effectively uncovers intrinsic redundancy to achieve robustness.

### 3.3. Curvature-aware Coloring Function

**Problem Formulation via Loss Geometry.** The core objective of our framework is to determine the optimal assignment function, $\alpha_{\text{layer}}$, which maps each layer $\ell$ to a shared basis from a candidate set $\mathcal{B} = \{B_k\}_{k=1}^K$, where each basis $B_k$ consists of a pair of low-rank matrices $U_k, V_k$. Rather than relying on heuristic weight distances, we formulate this selection based on the local geometry of the loss landscape. Let $W$ denote the original parameters and $\widehat{W}(\alpha_{\text{layer}})$ be the shared approximation. The perturbation introduced

by sharing is defined as $\delta = \widehat{W} - W$.

For a converged model, the first-order gradient vanishes. Consequently, the degradation in performance, $\Delta J$, is dominated by the second-order term in the Taylor expansion:

$$\Delta J \approx \frac{1}{2}\delta^\top H \delta, \tag{1}$$

where $H$ is the Hessian matrix. Geometrically, the level sets of this quadratic form define a multidimensional ellipsoid. The eigenvectors of $H$ determine the directions of the ellipsoid's axes. Specifically, eigenvectors associated with large eigenvalues correspond to high-curvature directions where the loss increases rapidly; these represent the ellipsoid's *minor axes*. In contrast, eigenvectors associated with small eigenvalues correspond to low-curvature (flat) directions where the loss is insensitive to perturbations; these represent the ellipsoid's *major axes*.

---

**Algorithm 1** Curvature-Aware Graph Coloring Algorithm

---

**Input:** Original weights $\{W_\ell\}$, candidate basis $\{B_b = (U_b, V_b)\}$, amplitude factor $\beta$, minor-axis count $t$.
**Output:** Assignment $\alpha_{\text{layer}}$, Layer Coefficients $\{S_\ell\}$.
1: Precompute minor-axis vectors $\{p_j^{(\ell)}\}$ for each layer $\ell$
2: **for** each layer $\ell$ **do**
3:     $\tau_\ell \leftarrow \beta \cdot \|W_\ell\|_F$
4:     // Select basis minimizing high-curvature energy, subject to total perturbation constraint
5:     $b^* \leftarrow \underset{b \in \{1..K\}}{\arg\min} \left\|\delta_\perp^{(b)}\right\|_2^2$   s.t.   $\left\|\delta_\parallel^{(b)}\right\|_2 \leq \tau_\ell$
6:     where
7:         $\delta^{(b)} = U_b(U_b^\top W_\ell V_b)V_b^\top - W_\ell$
8:         $\delta_\perp^{(b)} = \sum_{j=1}^t \langle p_j^{(\ell)}, \delta^{(b)}\rangle p_j^{(\ell)}$
9:     $\alpha_{\text{layer}}(\ell) \leftarrow b^*$
10:    $S_\ell \leftarrow U_{b^*}^\top W_\ell V_{b^*}$
11: **end for**
    **return** $\alpha_{\text{layer}}, \{S_\ell\}$

---

**Subspace Decomposition and Alignment.** To minimize accuracy loss, the perturbation $\delta$ must be aligned with the major axes (flat directions) of the loss ellipsoid. We formally decompose the perturbation into 2 orthogonal components:

> **High-curvature component** ($\delta_\perp$): The projection of $\delta$ onto the subspace spanned by the top-$t$ dominant eigenvectors (corresponding to the minor axes).

> **Low-curvature component** ($\delta_\parallel$): The projection onto the remaining flat subspace (corresponding to the major axes).

Since the eigenvalues in the high-curvature subspace are significantly larger, the optimization priority is to minimize the energy of $\delta_\perp$. However, we must also ensure the total perturbation remains within a reasonable range to validate the local quadratic approximation.

**Optimization Objective.** Based on this geometric insight, we define the coloring function as a constrained optimization problem. For each layer $\ell$, we select the basis $b^*$ that minimizes the projection error on the high-curvature directions, subject to a trust-region constraint on the total perturbation:

$$b^* = \underset{b \in \{1..K\}}{\arg\min} \left\|\delta_\perp^{(b)}\right\|_2^2 \quad \text{s.t.} \quad \left\|\delta_\parallel^{(b)}\right\|_2 \leq \tau_\ell, \tag{2}$$

where $\tau_\ell = \beta\|W_\ell\|_F$ defines the radius of the trust region, controlled by a hyperparameter $\beta$. This constraint prevents the shared parameters from drifting too far from the original weights, even along the major axes, ensuring the stability of the approximation.

### 3.4. Problem Solving and Parameter Reconstruction

The optimization problem from Eq. (2) is solved greedily, as detailed in Algorithm 1. This procedure operationalizes our graph-coloring framework: for each layer $\ell$ (a node), the algorithm iterates through every candidate basis $B_b$ (a color, generated following (Wang et al., 2025a)). It then selects the optimal color $b^*$ that minimizes the projection error onto the high-curvature subspace—our effective "coloring cost"—while respecting the trust-region constraint. The final output is the coloring function, $\alpha_{\text{layer}}$, which maps each layer to its optimal basis. This function does more than assign parameters; it uncovers latent structural properties. All layers sharing the same color (e.g., $\ell \mid \alpha_{\text{layer}}(\ell) = b^*$) form a set of interchangeable components, and the resulting partitioning of layers defines a non-trivial subgroup of the model's automorphism group, discovered automatically by our method. If no candidate basis satisfies the trust-region constraint $\|\delta_\parallel^{(b)}\|_2 \leq \tau_\ell$, we gradually relax $\tau_\ell$ by a factor of 1.2 until feasibility is reached, which guarantees convergence within two iterations in all experiments.

**Efficient Reconstruction.** This procedure of determining a 'color' for each layer directly yields a compressed model. Instead of storing full-rank weight matrices, for each layer $\ell$ we only store its assigned basis index (color), $\alpha_{\text{layer}}(\ell)$, and a small, layer-specific coefficient matrix $S_\ell$. The full weight matrix $\widehat{W}_\ell$ is then reconstructed on-the-fly only when needed during inference:

$$\widehat{W}_\ell = U_{b^*} S_\ell V_{b^*}^\top, \quad \text{where } b^* = \alpha_{\text{layer}}(\ell). \tag{3}$$

This low-rank formulation is the source of our method's efficiency. Parameter savings arise from storing the compact matrices $S_\ell$ instead of the full weights $W_\ell$. Simultaneously, inference is accelerated because the single, expensive full-matrix multiplication is replaced by two faster, smaller matrix multiplications.

*Table 1.* Our method's PPL (↓) and zero-shot (↑) performance under LLaMA-7B, following an SVD-based evaluation scheme on 3 language modeling datasets and 7 common-sense reasoning datasets. Ratio represents the compression rate.

| Ratio | Method | PTB↓ | C4↓ | WikiText-2↓ | Openb. | ARC_e | WinoG. | HellaS. | ARC_c | PIQA | MathQA | Average↑ |
|-------|--------|------|-----|-------------|--------|-------|--------|---------|-------|------|--------|----------|
| 0% | Original | 8.35 | 7.34 | 5.68 | 0.28 | 0.67 | 0.67 | 0.56 | 0.38 | 0.78 | 0.27 | 0.52 |
| 20% | SVD | 20306 | 18800 | 20061 | 0.14 | 0.27 | 0.51 | 0.26 | 0.21 | 0.53 | 0.21 | 0.31 |
| | FWSVD | 2152 | 1511 | 1727 | 0.15 | 0.31 | 0.50 | 0.26 | 0.23 | 0.56 | 0.21 | 0.32 |
| | ASVD | 16.55 | 15.93 | 11.14 | 0.25 | 0.53 | 0.64 | 0.41 | 0.27 | 0.68 | 0.24 | 0.43 |
| | SVD-LLM | 18.05 | 15.93 | 7.94 | 0.22 | 0.58 | 0.63 | 0.43 | 0.29 | 0.69 | 0.24 | 0.44 |
| | Basis Sharing | 17.35 | 15.03 | 7.74 | 0.28 | 0.66 | 0.66 | 0.46 | 0.36 | 0.71 | 0.25 | 0.48 |
| | **CGC(Ours)** | **16.54** | **13.88** | **7.07** | **0.29** | **0.66** | **0.69** | **0.46** | **0.37** | **0.71** | **0.25** | **0.49** |
| 30% | SVD | 17210 | 20871 | 13103 | 0.13 | 0.26 | 0.51 | 0.26 | 0.21 | 0.54 | 0.22 | 0.30 |
| | FWSVD | 11058 | 7240 | 20127 | 0.17 | 0.26 | 0.49 | 0.26 | 0.22 | 0.51 | 0.19 | 0.30 |
| | ASVD | 70 | 41 | 51 | 0.18 | 0.43 | 0.53 | 0.37 | 0.25 | 0.65 | 0.21 | 0.38 |
| | SVD-LLM | 29.44 | 25.11 | 9.56 | 0.20 | 0.48 | 0.59 | 0.40 | 0.26 | 0.65 | 0.22 | 0.40 |
| | Basis Sharing | 29.12 | 22.46 | 9.25 | 0.27 | 0.63 | 0.63 | 0.40 | 0.30 | 0.68 | 0.24 | 0.45 |
| | **CGC(Ours)** | **27.65** | **21.89** | **9.13** | **0.28** | **0.65** | **0.66** | **0.41** | **0.33** | **0.69** | **0.24** | **0.47** |
| 40% | SVD | 59977 | 47774 | 52489 | 0.15 | 0.26 | 0.52 | 0.26 | 0.22 | 0.53 | 0.20 | 0.30 |
| | FWSVD | 20990 | 12847 | 18156 | 0.16 | 0.26 | 0.51 | 0.26 | 0.22 | 0.53 | 0.21 | 0.30 |
| | ASVD | 3292 | 1109 | 1407 | 0.13 | 0.28 | 0.48 | 0.26 | 0.22 | 0.55 | 0.19 | 0.30 |
| | SVD-LLM | 63.75 | 49.83 | 13.11 | 0.19 | 0.42 | 0.58 | 0.33 | 0.25 | 0.60 | 0.21 | 0.37 |
| | Basis Sharing | 55.78 | 41.28 | 12.39 | 0.22 | 0.52 | 0.61 | 0.35 | 0.27 | 0.62 | **0.23** | 0.40 |
| | **CGC(Ours)** | **52.47** | **39.78** | **12.16** | **0.23** | **0.55** | **0.62** | **0.36** | **0.28** | **0.64** | **0.23** | **0.42** |
| 50% | SVD | 87227 | 79815 | 131715 | 0.16 | 0.26 | 0.50 | 0.26 | 0.23 | 0.52 | 0.19 | 0.30 |
| | FWSVD | 28321 | 23104 | 24391 | 0.12 | 0.26 | 0.50 | 0.26 | 0.23 | 0.53 | 0.20 | 0.30 |
| | ASVD | 47690 | 27925 | 15358 | 0.12 | 0.26 | 0.51 | 0.26 | 0.22 | 0.52 | 0.19 | 0.30 |
| | SVD-LLM | 150.58 | 118.57 | 23.97 | 0.16 | 0.33 | 0.54 | 0.29 | 0.23 | 0.56 | 0.21 | 0.33 |
| | Basis Sharing | 126.35 | 88.44 | 19.99 | 0.18 | 0.42 | 0.57 | **0.31** | 0.23 | 0.58 | 0.22 | 0.36 |
| | **CGC(Ours)** | **117.23** | **79.01** | **18.95** | **0.20** | **0.45** | **0.60** | **0.31** | **0.24** | **0.60** | **0.22** | **0.37** |

## 4. Experiments

We structure the experiments into three parts: **(1)** Downstream performance across Large Language Models (LLMs) and Vision Transformers to verify the method's generalizability and fidelity retention; **(2)** CGC parameter sharing analysis, demonstrating the reduction in shared parameters and the curvature effectiveness; and **(3)** Computational benchmarks, highlighting enhanced inference efficiency.

### 4.1. Experiment Configuration

#### 4.1.1. MODEL

**Visual Transformers.** For visual Transformers, we evaluate the Swin-Transformer (Liu et al., 2021) on ImageNet (Krizhevsky et al., 2012) and transfer it to downstream tasks such as CIFAR (Krizhevsky et al., 2009).

**Large Language Model.** For large language models (LLMs), we conduct experiments on multiple architectures, including the Qwen3 family(Qwen et al., 2025), LLaMA (Touvron et al., 2023) family (LLaMA-7B, LLaMA-13B, LLaMA-30B, LLaMA2-7B, LLaMA3.1-8B, LLaMA3.2

3B), OPT-6.7B (Zhang et al., 2022), and Mistral-7B.

#### 4.1.2. DATASET AND CONFIGURATION

**Dataset.** Our evaluation encompasses 3 language modeling datasets: WikiText-2 (Merity et al., 2016), Penn Treebank (PTB) (Marcinkiewicz, 1994), and C4 (Raffel et al., 2020). Additionally, we assess performance on seven reasoning datasets: OpenbookQA (Banerjee et al., 2019), WinoGrande (Sakaguchi et al., 2019), HellaSwag (Zellers et al., 2019), PIQA (Bisk et al., 2020), MathQA (Amini et al., 2019), ARC-easy, and ARC-challenge (Clark et al., 2018). All reasoning tasks are evaluated under zero-shot settings using the LM-Evaluation-Harness framework to ensure consistent and reproducible results.

**Configuration.** All models are implemented using Hugging Face transformers. LLaMA-30B is implemented in FP16 precision, while all other models use FP32. For sharing, we follow the Basis Sharing. Hessian terms and eigenvectors are approximated using Hessian-vector products combined with the Lanczos algorithm. The data for the methods listed in the table are from publicly published papers. The number of short-axis eigenvalues $t$ is set to 550, and the perturbation

*Table 2.* Comparison of our method's PPL (↓) performance on LLaMA2-7B with the baseline under different compression ratios.

| Ratio | Method | PTB↓ | C4↓ | WikiText-2↓ |
|-------|--------|------|-----|-------------|
| 0% | Original | 7.29 | 7.29 | 5.47 |
| 20% | Basis Sharing | 60 | 15.3 | 7.77 |
| | **CGC(Ours)** | **54.53** | **14.9** | **7.57** |
| 30% | Basis Sharing | 97.4 | 23.86 | 9.69 |
| | **CGC(Ours)** | **88.33** | **23.17** | **9.52** |
| 40% | Basis Sharing | 195.95 | 43.89 | 13.62 |
| | **CGC(Ours)** | **175.55** | **41.49** | **13.48** |
| 50% | Basis Sharing | 509.3 | 98.92 | 21.3 |
| | **CGC(Ours)** | **371.75** | **88.27** | **20.16** |

*Table 3.* Scalability results of the PPL(↓) for larger-scale LLMs on WikiText-2.

| Model | LLaMA-7B | LLaMA-13B | LLaMA-30B |
|-------|----------|-----------|-----------|
| SVD | 20061 | 946.31 | 54.11 |
| FWSVD | 1630 | OOM | OOM |
| SVD-LLM | 7.94 | 6.61 | 5.63 |
| Basis Sharing | 7.74 | 6.51 | 5.47 |
| **CGC(Ours)** | **7.07** | **6.21** | **5.33** |

*Table 4.* Perplexity comparison for LLaMA-7B between our method and Basis Sharing under extreme compression ratios on C4 and WikiText-2.

| Dataset | Basis Sharing | Ours | Ratio |
|---------|---------------|------|-------|
| C4 | 651.8314 | **603.4069** | 70% |
| WikiText-2 | 136.8194 | **125.0952** | |
| C4 | 2465.999 | **995.33** | 80% |
| WikiText-2 | 624.0834 | **424.8948** | |

amplitude $\beta$ is set to 5e-2; these hyperparameter choices will be justified in the ablation studies. Following (Wang et al., 2025a), we construct $\mathcal{B} = \{B_k\}_{k=1}^K$ by aggregating layer-wise weight matrices $\{W_\ell\}_{\ell \in L}$ within each parameter type (Q, K, V, FFN, etc.). Specifically, for each type we perform SVD on the concatenated matrices: $\tilde{W} = [W_1; W_2; \ldots]$, retain the top-$r$ singular vectors as the shared basis $B_k$. The number of basis $K$ is determined by the target compression ratio $\rho$. Experiments were implemented in PyTorch on 2 NVIDIA A800 GPUs.

### 4.2. Downstream Performance Experiments

#### 4.2.1. LARGE LANGUAGE MODEL

**Comparison Results.** As shown in Table 1, across compression ratios, our method consistently outperforms all SVD-based baselines on both language modeling and zero-shot reasoning. Even under high compression, the model maintains stable per perplexity and accuracy, indicating that the proposed major-axis alignment and perturbation control effectively preserve key representational structures. On LLaMA-7B, our approach achieves lower perplexity and higher average zero-shot scores across all datasets, showing clear advantages in both linguistic coherence and reasoning generalization. The gap further widens at higher compression, confirming that aligning shared subspaces with low-curvature Hessian directions improves robustness.

Similar trends appear on LLaMA2-7B in Table 2, where our method achieves lower PPL than Basis-Sharing across all tested ratios. This suggests that the proposed geometric basis selection and adaptive amplitude modulation not only minimize distortion from rank reduction but also preserve global semantic consistency. These results demonstrate that CGC scales effectively to large models, preserving fluency and reasoning ability under strong compression.

**Larger-scale LLMs.** To verify adaptability on larger models, we extend experiments to LLaMA-7B, 13B, and 30B. As shown in Table 3, existing methods (FWSVD) fail to

scale due to high memory cost(OOM), while ours achieves the lowest perplexity across all sizes.

**Extreme compression.** At extreme compression ratios of 70–80% for LLaMA-7B model, where performance degradation is inevitable for any method, our approach demonstrates significant relative gains on identical setting. Our method consistently outperforms Basis Sharing with lower perplexity on both C4 and WikiText-2 datasets (Table 4). Even at 80% compression, it reduces perplexity by over 50% on C4, showcasing the representational stability.

**Diverse LLM Architectures.** We evaluate CGC across a wide range of architectures to demonstrate its universality. First, as shown in Table 5, our method outperforms existing compression techniques (e.g., ASVD, Basis Sharing) on OPT-6.7B, Qwen3-4B, and Mistral-7B, achieving the lowest perplexity under a 20% compression ratio.

Furthermore, to verify robustness, Table 6 extends the evaluation to more advanced architectures. The results indicate that CGC maintains performance close to the original model across diverse families (Qwen, LLaMA3) and scales effectively from 0.6B to 8B parameters, showing strong generalization capabilities without model-specific tuning.

#### 4.2.2. VISION TRANSFORMER MODEL

Table 7 reports results on ImageNet using Swin-Transformer and DeiT backbones at same compression ratio. Compared with SVD-based baselines such as GFM (Yu & Wu, 2023), FiPS, and LossFac (Zhang et al., 2024), our method attains the smallest accuracy drop at a comparable compression ratio (28%). The post-sharing Top-1 remains nearly identical to the original model, showing that the proposed alignment mechanism effectively preserves representational capacity

*Table 5.* Perplexity comparison on WikiText-2 across diverse LLM architectures under a 20% compression ratio.

| Method | Qwen3-4B | OPT-6.7B | Mistral-7B |
|---|---|---|---|
| ASVD | - | 82 | 10.21 |
| SVD-LLMv2 | - | 13.46 | - |
| Basis Sharing | 15.98 | 11.79 | 7.57 |
| CGC(Ours) | **15.02** | **11.68** | **7.49** |

*Table 6.* Generalization evaluation (PPL) across different, more advanced architectures and datasets.

| Models | Qwen3 0.6B | Llama3.2 3B | Llama-3.1 8B |
|---|---|---|---|
| **WikiText-2** | | | |
| Original | 21.19 | 7.82 | 7.23 |
| CGC(Ours) | 23.45 | 9.11 | 8.5 |
| **C4** | | | |
| Original | 30.19 | 11.3 | 11.62 |
| CGC(Ours) | 36.34 | 18.31 | 14.02 |

under strong compression. These results confirm that CGC generalizes well beyond language models, adapting effectively to vision backbones with hierarchical structures.

*Table 7.* Performance comparison of our method and existing sharing methods for Vision-Transformer on ImageNet.

| Model | Method | Top1 | Top1-Share | Top1-Drop |
|---|---|---|---|---|
| Swin-L | AAFM | 86.25 | 85.73 | -0.52 |
| | GFM | 86.25 | 85.83 | -0.42 |
| | FiPS | 86.24 | 86.21 | -0.03 |
| | LossFac | 86.23 | 86.19 | -0.04 |
| | **Ours** | 86.24 | 86.23 | **-0.01** |
| DeiT-B | PELA | 81.80 | 79.46 | -2.34 |
| | **Ours** | 81.87 | 81.84 | **-0.03** |

**Transferring Ability.** In Table 8, we transfer the shared model to 2 downstream tasks, including CIFAR-10/100. Consistent with the results on ImageNet, our method achieves accuracy on par with the original model (Dosovitskiy, 2020; Touvron et al., 2021) on these downstream tasks. This indicates that parameter sharing preserves the model's generalization capability.

**Comparison with Quantization Methods.** Although CGC is primarily designed as a structured parameter-sharing method, we further compare it with representative quantization methods to better position it among different compression paradigms. We report the results under very low memory budgets on LLaMA2-7B/WikiText-2. Compared with PB-LLM (Shang et al., 2023) and GPTQ (Frantar et al., 2022), CGC achieves lower perplexity while using a slightly smaller memory budget. Specifically, CGC obtains a PPL of 98.21 with 1.8GB memory, whereas PB-LLM and GPTQ obtain PPLs of 104.83 and 110.99 with 1.9GB memory, re-

*Table 8.* Comparison of transfer learning results between multiple visual models and sharing model at different compression rates. Drop indicates the magnitude of performance drop.

| | Model | Acc / Share(%) | Drop | F1 / Share(%) | Drop | Ratio |
|---|---|---|---|---|---|---|
| **CIFAR-10** | Swin-L | 97.70/97.41 | -0.39 | 97.67/97.15 | -0.52 | 20% |
| | Swin-B | 90.81/91.00 | +0.20 | 90.77/91.44 | +0.67 | 20% |
| | DeiT-B | 92.90/91.90 | -1.00 | 92.87/91.90 | -0.97 | 30% |
| | Swin-B | 90.81/91.36 | +0.45 | 90.77/91.35 | +0.58 | 30% |
| **CIFAR-100** | Swin-L | 82.82/81.72 | -1.10 | 81.52/80.66 | -0.86 | 20% |
| | Swin-B | 67.39/68.05 | +0.66 | 65.35/66.06 | +0.71 | 20% |
| | DeiT-B | 72.80/70.30 | -2.50 | 71.33/69.56 | -1.78 | 30% |
| | Swin-B | 67.39/67.47 | +0.08 | 65.35/65.14 | +0.21 | 30% |

spectively. These results suggest that CGC remains effective in ultra-low-memory settings. We emphasize that this comparison is intended to position CGC relative to quantization-based compression, rather than to claim a definitive ranking across different compression paradigms.

### 4.3. Parameter Sharing Results Analysis.

**Visualizing Symmetry-Induced Structure.** In Figure 3(a), our method successfully represents a 32-layer network using only 12 shared bases, significantly fewer than the 16 used by Basis Sharing (BS) and the 32 in the non-sharing baseline. Figure 3(b) reveals the specific mapping on LLaMA-7B. Crucially, our method discovers non-local shared-basis compatibility that heuristic methods miss. For instance, layers 15, 17, and 18 are assigned the same color (basis) despite being non-adjacent. This suggests that CGC breaks the limitation of "neighbor-only" sharing by identifying layers for which a common low-rank basis induces small curvature-weighted perturbation.

Importantly, same-color assignment does not imply that these layers are functionally equivalent. Layers assigned to the same color share the basis factors $(U, V)$, but each layer still retains its own coefficient matrix $S_\ell$. Therefore, the reconstructed weights can still represent layer-specific transformations. Same-color groups should be interpreted as basis-compatible components rather than interchangeable functional modules. From a structural perspective, the partition induced by $\alpha_{\text{layer}}$ can be viewed as a basis-sharing symmetry pattern. We quantify this pattern using $|Aut(G)| = \prod_b |L_b|!$. As detailed in Appendix C, larger sharing groups generally correspond to lower reconstruction error, suggesting that CGC finds compact sharing structures compatible with the model's local loss geometry.

**Validating Curvature-aware Effectiveness.** To verify whether the performance gain stems from our curvature-based guidance or merely the basis generation process, Figure 4 validates the geometric effectiveness of our curvature-aware coloring. We compare the L2 reconstruction error of

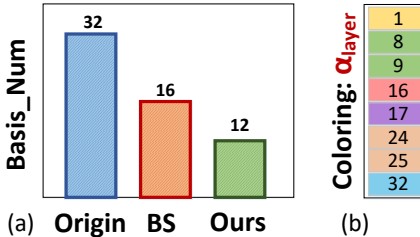

*Figure 3.* (a) Comparison of the number of basis in our method (32 layers represented by only 12 basis) with (Wang et al., 2025a). (b) Specific coloring scheme $\alpha_{\text{layer}}$ of our method when compressing LLaMA 7B by 50% (same color indicates shared basis).

layers 7–10 across different parameter modules (Gate, K, Q, Up, V) under two strategies: heuristic adjacent grouping and our CGC, with the basis generation algorithm held constant. CGC yields consistently lower errors across all modules. This demonstrates that by aligning sharing perturbations with the Hessian's major axes, our method minimizes post-sharing distortion. This result confirms that the improvement is not solely due to the basis generation, but is directly attributable to the curvature-aware selection strategy, which identifies high-affinity layers that minimize distortion more effectively than heuristic proximity.

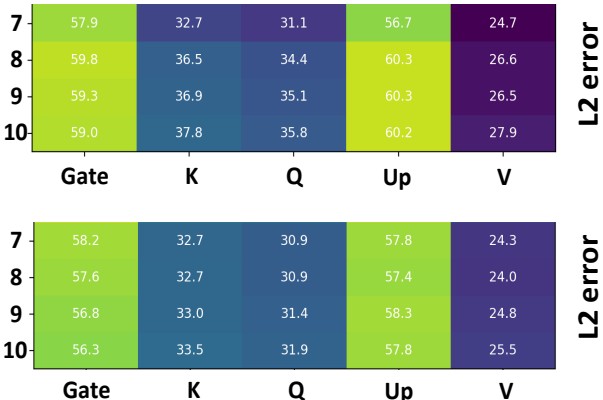

*Figure 4.* Comparison of L2 error caused by different coloring functions (top: adjacent, bottom: CGC), including the difference between the weights after sharing in layers 7-10 and the weights of the original standard model.

### 4.4. Computational Performance Experiments

**Inference Efficiency on Real Hardware.** Table 9 summarizes the inference efficiency of our method across three representative models, evaluated on a single NVIDIA A800 GPU with batch size 512 and sequence length 32. Across all models, our approach consistently reduces both parameter count and MACs by around 40–50%, leading to nearly 45% lower latency and up to 70% higher throughput compared with the original model. These results indicate that our method not only maintains model accuracy but also

brings substantial runtime benefits, showing strong generalization across different architectures and demonstrating its practicality for large-scale deployment.

**Hessian and Algorithm Time.** We conducted a detailed breakdown of the computational overhead introduced by Hessian estimation, computed layer-wise for each linear module using a standard calibration subset. For a typical setup (batch size 1, context length 512, 20 iterations), the process involves two main stages:

Hessian Eigenvalue Computation: Calculating the top-550 eigenvalues takes approximately 0.93 hours.

Energy Minimization: The subsequent high-curvature energy minimization requires 0.4 hours.

This runtime is consistent with other SOTA Hessian-based methods. As a one-time offline cost, this "curvature budget" is highly efficient, trading marginal preparation time for superior fidelity without increasing inference latency.

Dynamic Tying (Hay & Wolf, 2024) takes around **13.8 hours with PPL:49.37** on WikiText-2. CGC demonstrates significantly faster efficiency while achieving lower PPL (**42.30**). During deployment, our method maintains lower PPL while keeping inference time comparable to Basis Sharing.

### 4.5. Ablations

**The Number of Short-Axis.** Figure 5(a) examines the effect of the short-axis count $t$ in Algorithm. Increasing $t$ refines the estimation of high-curvature directions, allowing more accurate projection of perturbations onto the flat subspace. Consequently, perplexity decreases steadily, but computation time grows nearly linearly due to higher projection cost. This confirms that larger $t$ improves the curvature fidelity of the alignment but at the expense of efficiency.

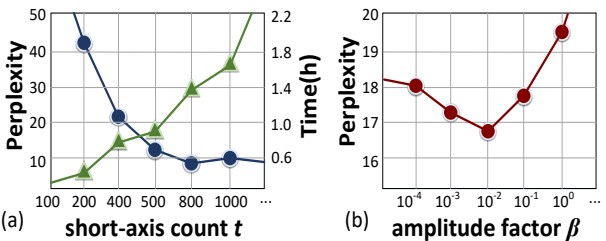

*Figure 5.* Ablations. (a) As the number of minor axes increases, perplexity consistently decreases, though computational burden increases. (b) When the amplitude factor increases, excessive perturbation leads to a sharp surge in perplexity.

**Robustness to Hessian Approximation Errors.** To directly evaluate whether CGC relies on an overly accurate Hessian estimate, we perturb the estimated short-axis eigenspace and rerun the assignment procedure. Table 10 shows that the perplexity degrades smoothly as the perturbation level

*Table 9.* Inference efficiency of our method on real hardware.

| Model | Params.(B) | MACs(B) | Latency(s) | Throughput(t/s) |
|---|---|---|---|---|
| LLaMA2-7B | 6.74B | 6.61B | 13.21 | 1338.37 |
| **Ours** | **3.50** ↓48.1% | **3.94** ↓40.4% | **7.06** ↓46.6% | **2152.92** ↑60.9% |
| LLaMA-7B | 6.74B | 6.61B | 13.27 | 1331.88 |
| **Ours** | **3.99** ↓40.8% | **3.94** ↓40.4% | **7.38** ↓44.4% | **2084.30** ↑56.5% |
| Mistral-7B | 7.24B | 7.11B | 14.61 | 1248.48 |
| **Ours** | **3.75** ↓48.2% | **3.99** ↓43.9% | **7.93** ↓45.7% | **2135.37** ↑71.0% |

increases. This suggests that CGC only requires a stable approximation of the dominant high-curvature subspace, rather than exact recovery of the full Hessian.

*Table 10.* Robustness to perturbations of the estimated Hessian eigenspace on LLaMA-7B/WikiText-2.

| Perturbation level | 0 | 0.1 | 0.3 | 0.5 | 0.8 |
|---|---|---|---|---|---|
| PPL ↓ | 7.07 | 7.07 | 7.22 | 7.43 | 7.86 |

**The Amplitude Factor.** Figure 5(b) investigates the amplitude factor $\beta$, which controls the trust-region radius in alignment. Small $\beta$ overly restrict perturbations, preventing sufficient movement along flat directions and leading to underfitting. As $\beta$ increases to around $10^{-2} - 10^{-1}$, moderate perturbation energy improves alignment and yields the lowest perplexity. Beyond this range, excessive amplitude breaks the local quadratic assumption and injects noise into sensitive directions, sharply degrading performance.

**The First-Order Term.** To verify the validity of the second-order approximation in Eq.1, we evaluate the first-order contribution in the Taylor expansion on ViT. We compute the ratio $c = 2|\nabla_W \mathcal{J}(W)^\top \delta|/|\delta^\top H \delta|$, for each layer and each sampled perturbation $\delta$ obtained from the sharing process. Empirically, $c < 0.04$ ($c < 0.1$ in LLaMA) for 90% of layers, indicating that the first-order term is negligible compared with the second-order curvature term. This phenomenon arises because the model is already well optimized—the gradient norm $\|\nabla_W \mathcal{J}\|$ is close to zero—making the first-order term vanish at convergence. Consequently, the loss change is dominated by the second-order, which validates the assumption used in Eq.1 and supports our sharing strategy.

## 5. Conclusion

This work presents Curvature-aware Graph Coloring (CGC), a structured framework that reformulates parameter sharing by aligning layer subspaces with low-curvature Hessian directions. Grounded in symmetry insights, CGC replaces heuristic rules with a principled, training-free strategy that offers a unified view of cross-layer redundancy. Experi-

ments on vision and language models confirm that CGC achieves superior compression–accuracy trade-offs compared to both heuristic and SVD-based baselines.

## Acknowledgments

This work was supported by the National Key Research and Development Program of China under Grant 2024YFB4505602, the National Natural Science Foundation of China under Grant 62372432.

This work was also supported in part by the Major Key Project of PCL under Grant PCL2024A06 and PCL2025A10, and in part by the Shenzhen Science and Technology Program under Grant RCJC20231211085918010. This work was also supported in part by NSFC (62476070).

## Impact Statement

The goal of this paper is to advance the field of machine learning compression. However, our work has no potential negative social impact and is only intended to provide a reference for future research in this area.

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

## Appendix A. Justification of the Alignment Objective

We provide a concise justification for the curvature-aware assignment objective used in Eq. (2) and Algorithm 1. Let $W_\ell$ be the original weight of layer $\ell$ and let $\widehat{W}_\ell$ be its shared approximation. The sharing-induced perturbation is

$$\delta_\ell = \widehat{W}_\ell - W_\ell.$$

Around a well-optimized solution, the first-order term is small, and the loss change can be approximated by the second-order Taylor expansion:

$$\Delta \mathcal{L}_\ell \approx \frac{1}{2} \delta_\ell^\top H_\ell \delta_\ell,$$

where $H_\ell$ is the layer-wise Hessian.

Let $\{p_j^{(\ell)}\}_{j=1}^t$ denote the top-$t$ eigenvectors of $H_\ell$, corresponding to the high-curvature directions. We decompose the perturbation into

$$\delta_{\ell,\perp} = \sum_{j=1}^t \langle p_j^{(\ell)}, \delta_\ell \rangle p_j^{(\ell)}, \quad \delta_{\ell,\|} = \delta_\ell - \delta_{\ell,\perp}.$$

Here, $\delta_{\ell,\perp}$ is the high-curvature component, while $\delta_{\ell,\|}$ lies in the remaining low-curvature subspace.

Since perturbations along high-curvature directions induce larger loss changes, CGC selects the candidate shared basis that minimizes the high-curvature projection energy while keeping the total perturbation within a trust region:

$$\min_{B_b} \|\delta_{\ell,\perp}(B_b)\|_2^2 \quad \text{s.t.} \quad \|\delta_\ell(B_b)\|_2 \leq \tau_\ell.$$

This is the objective implemented in Algorithm 1. The trust-region constraint prevents the approximation from drifting too far from the original weight, while the projection term encourages the sharing noise to align with low-curvature directions of the local loss landscape.

## Appendix B. Basis Compatibility and Coloring Interpretation

CGC uses a coloring-inspired formulation to describe global basis assignment. Each color denotes a shared low-rank basis. Layers assigned to the same color share the basis factors $(U_b, V_b)$, but still keep their own layer-specific coefficient matrices $S_{\ell,b}$. Therefore, same-color assignment should be interpreted as compatibility with a shared basis, rather than functional equivalence between layers.

The role of curvature is to evaluate whether a candidate basis induces a safe sharing perturbation for each layer. Specifically, CGC assigns a layer to a basis when the induced perturbation has small projection onto the layer's

high-curvature subspace and satisfies the trust-region constraint. Thus, the coloring view specifies which basis is reused, while the curvature-aware objective specifies which assignments are locally stable.

## Appendix C. More Experiments

**Additional results on CIFAR-100.** We further evaluate ViT-B on CIFAR-100 to examine whether the proposed curvature-aware basis assignment remains stable on downstream vision tasks. As shown in Table 11, CGC preserves both accuracy and F1 score under moderate compression. At the $1.2\times$ compression setting, the shared model achieves slightly higher accuracy than the original model, while the F1 score remains almost unchanged. At the $1.3\times$ compression setting, the performance drop is still small. These results provide additional evidence that CGC maintains stable generalization behavior beyond the main ImageNet experiments.

**Comparison with pruning and sharing baselines.** We also compare CGC with two representative compression methods: ShortGPT-BI, a pruning-based method, and FoldGPT, a parameter-sharing method. As shown in Table 12, CGC achieves a PPL of 9.52 at a 30% compression ratio, while FoldGPT and ShortGPT-BI obtain PPLs of 18.5 and 35.96 at 27% compression, respectively. Although the compression ratios are not exactly identical, this comparison provides additional evidence that CGC preserves language modeling performance well under strong compression. This result also helps position CGC relative to other compression paradigms, while the main comparisons in the paper focus on low-rank and parameter-sharing baselines.

**Sharing-pattern analysis.** We further characterize the compactness of the sharing pattern discovered by CGC. For a coloring assignment $\alpha : \mathcal{L} \to \mathcal{B}$, let

$$\mathcal{L}_b = \{\ell \mid \alpha(\ell) = b\}$$

denote the set of layers assigned to basis $b$. We report the group-size metric

$$\log_2 |\mathcal{G}(\alpha)| = \sum_b \log_2(|\mathcal{L}_b|!).$$

A larger value indicates that more layers are grouped under shared bases. Importantly, this metric is used only to describe the compactness of the basis-sharing pattern, not to claim that same-color layers are functionally identical.

As shown in Table 13, CGC achieves a larger group-size metric than Basis Sharing while also obtaining lower perplexity. Specifically, CGC uses 12 bases and achieves a PPL of 79.01, whereas Basis Sharing uses 16 bases and obtains

*Table 11.* Performance of CGC on ViT-B evaluated on CIFAR-100. The method maintains accuracy and F1(%) under moderate compression. Positive changes indicate improvements after sharing.

| Model | Ratio | Acc | Acc-share | Acc-Change | F1 | F1-share | F1-Change |
|-------|-------|-----|-----------|------------|-----|----------|-----------|
| ViT-B | 1.3× | 75.40% | 74.70% | -0.70 | 73.89 | 73.57 | -0.32 |
| ViT-B | 1.2× | 75.40% | 75.50% | +0.10 | 73.89 | 73.84 | -0.04 |

*Table 12.* Comparison of compression methods on LLaMA2-7B using PPL as the evaluation metric.

| Method | Compression Ratio | PPL |
|--------|-------------------|-----|
| FoldGPT | 27% | 18.5 |
| ShortGPT-BI | 27% | 35.96 |
| Ours | **30%** | **9.52** |

a PPL of 88.44. This suggests that curvature-aware assignment can identify more compact sharing structures without sacrificing model quality. The result supports our claim that CGC discovers effective shared-basis compatibility beyond simple adjacent-layer sharing.

*Table 13.* Sharing-pattern analysis. The logarithmic group-size metric $\log_2 |\mathcal{G}(\alpha)| = \sum_b \log_2(|\mathcal{L}_b|!)$ measures the compactness of the basis-sharing pattern.

| Method | $|\mathcal{B}|$ | $\log_2 |\mathcal{G}(\alpha)| \uparrow$ | PPL $\downarrow$ | Ratio |
|--------|-----|--------------------------|--------|-------|
| SVD-LLM | 32 | 0 | 118.57 | 50% |
| Basis Sharing | 16 | 16.0 | 88.44 | 50% |
| Ours | 12 | **27.1** | **79.01** | 50% |

