# OpenReview forum: "Rethinking Parameter Sharing as Graph Coloring for Structured Compression"
_ICML.cc/2026/Conference — ICML 2026 regular_

### Official Review · Reviewer_3S7r · 2026-02-28

**Soundness:** 3
**Presentation:** 3
**Significance:** 2
**Originality:** 3
**Overall Recommendation:** 4
**Confidence:** 3

**Summary:**

This paper proposes Curvature-aware Graph Coloring (CGC), a model compression method that reformulates cross-layer parameter sharing as a graph coloring problem guided by loss landscape geometry. Instead of relying on singular value decomposition (SVD) or low-rank approximation based sharing strategies, CGC relies on second-order information from the Hessian matrix to group layers with similar curvature profiles. By aligning sharing-induced perturbations with low-curvature (flat) directions of the loss surface, the method minimizes performance degradation while enabling flexible, global cross-layer low-rank basis sharing. Experiments on large language models (e.g., LLaMA-7B and vision transformers (e.g., Swin Transformer) demonstrate that CGC achieves 28%–50% parameter compression with minimal or negligible accuracy loss.

**Compliance With Llm Reviewing Policy:**

Affirmed.

**Final Justification:**

The author resolved most of my concerns. Score has already been raised.

**Key Questions For Authors:**

1. For Table 1: What are the baseline performance numbers?
2. For Table 5: Which dataset were used?
3. For Table 6: Which evaluation metric is used in the table?
4. For 4.4, the computational performance were conduced only with batch size of 512, what about small batch sizes like 8, 16, 32?

**Limitations:**

No.

**Strengths And Weaknesses:**

Strength:
1. The idea of leveraging second-order information from the Hessian matrix to group layers with similar curvature profiles for model compression is novel. The authors innovate a curvature-aware graph coloring method to assign shared parameters to different layers.
2. The proposed method naturally supports flexible arbitrary cross-layer parameter sharing.
3. In the evaluation, the authors prove that the proposed method, Curvature-aware Graph Coloring (CGC), works with multiple different models including LLaMA-7B, LLaMA2-7B, LLaMA3.2 3B, OPT-6.7B, etc. and vision transformer models.

Weakness:
1. Mixture-of-Experts (MoE) architectures dominate many recent LLM developments. However, the applicability of CGC to MoE-based models is not discussed in the paper.
2. Quantized LLMs have become increasingly common, and many newly released models are distributed with pre-quantized weights. Whether CGC can work with quantized models (fp8 or mxfp4) are not discussed. (The experiments were conducted mostly on fp32 models)
3. Most of experiments are conducted on very old models like LLaMA-7B and LLaMA2-7B. Would like to see more experiments on most recent models like Qwen3 models, OpenAI GPT-OSS models, etc.

---

> ### Author Rebuttal · Authors · 2026-03-30
>
> Thank you for the careful review and constructive suggestions.
>
> We are pleased that the reviewer recognizes **the novelty of our work**, its **flexible cross-layer sharing capability**, and its **validation across multiple models**.
>
> **Q1: Applicability to newer models and MoE architectures.**
>
>  **A1:** We would first like to clarify that the current submission does not only include earlier LLaMA models; as specified in **Sec. 4.1.1**, we already evaluate several newer **dense LLMs** in the original paper, including **Qwen3 [2025], LLaMA3.1, and LLaMA3.2 [2024]**, with results reported in **Tables 5–6** (Qwen3) and **Table 6** (LLaMA3.1/LLaMA3.2).
>
> Regarding **MoE**, we agree that it has become an important architecture in recent LLMs, and thus its applicability is worth discussing. In practical applications, **MoE typically constitutes a more scenario-specific compression setting** *[Lu et al., 2024, Chen et al., 2025, Li et al., 2025b]*: compression is often performed at the expert level and is coupled with the routing mechanism, making it more suitable to be treated as a separate setting for validating applicability.
>
> Nevertheless, the core **curvature-aware assignment** principle underlying **CGC is not restricted to a specific dense architecture**.
>
> Specifically, we follow the MoE compression evaluation protocols of ***NAEE** [Lu et al., 2024]* and ***HC-SMoE** [Chen et al., 2025]*. In implementation, we perform structured sharing over atomic expert units inside MoE FFN experts. The initial candidate bases are provided by *MoE-SVD [Li et al., 2025b]*, after which CGC is applied for cross-layer sharing.
>
> |Model | Method | Ratio | ARC-e| ARC-c| HellaSwag | PIQA     |
> |-|-|-|-| -| -| -|
> | GPT-OSS 20B   | Origin   | 0 | 76.9| 45.2| 40.7| 76.0 |
> || NAEE| 0.2| 70.3 | 35.3 | 34.5 | 70.2|
> || **Ours** | 0.2| **72.6** | **35.3** | **37.5**  | **71.8** |
> | Qwen3-30B-A3B | Origin   | 0| 79.4| 54.0 | 60.2| 79.0     |
> || HC-SMoE  | 0.25  | 64.0     | 35.1 | 40.2 | 59.2|
> || **Ours** | 0.25  | **69.3** | **38.3** | **44.8**  | **62.8** |
>
> These results suggest that although CGC is not tailored for MoE, its **curvature-aware assignment mechanism transfers well to the MoE setting**.
>
> **Q2: Applicability to quantized models.**
>
>  **A2:** We further supplement experiments on **quantized models** to verify the applicability of CGC in low-bit weight spaces.
>
> Overall, we follow the low-precision evaluation settings of *FP8-AMAX* and the *MXFP4 benchmark*.  All quantized checkpoints are constructed with support from *HuggingFace*, *llmcompressor*, and *intel/neural-compressor*, using *RTN quantization*. Importantly, sharing is performed directly in the quantized weight space, rather than reusing the sharing results obtained from the FP16 model.
>
> | Qwen3-8B|WikiText-2 PPL↓ |
> | -| -|
> | Full-prec| 9.71|
> | *FP8 + Ours* | **12.82**|
> | *MXFP4 + Ours* | **17.41** |
> | *INT4 + Ours*| **19.99** |
>
> On *Qwen3-8B*, CGC remains **effective under multiple numerical formats.**
>
> We also measured **the offline overhead in the quantized setting.** On Qwen3-8B, CGC is faster on 4-bit weights than in full precision **(0.98h vs. 1.64h)**, suggesting that **quantization does not hinder applicability and can even reduce preprocessing cost.** Empirically, this is because fewer short axes were sufficient to estimate the curvature information well in our quantized experiments.
>
> |Qwen3-8B| Runtime↓|
> |-|-|
> |Ours (FP)|1.64h|
> |Ours (INT4)|**0.98h**|
>
> We will clarify this observation in the revised version.
>
> **Q3: Clarifications for Tables 1/5/6 and small-batch efficiency.**
>
> **A3:** Thank you for pointing this out. We agree that the current manuscript is not sufficiently clear on these tables. We clarify them below:
>
> 1.The baseline in **Table 1** corresponds to **“0% Original”**;
>
> 2.**Table 5** reports **PPL on WikiText-2**;
>
> 3.**Table 6** reports **PPL on WikiText-2 and C4**;
>
> We also supplement real-hardware results for **batch sizes 8/16/32** (in addition to batch size 512 in Sec. 4.4):
>
> | Batch size | Original latency (s)↓| CGC latency(s)↓| Latency change | Original throughput (tokens/s)↑| CGC throughput (tokens/s)↑| Throughput change |
> |-|-|-|-|-|-|-|
> |8| 0.2167| **0.1309**| -39.59% | 1151.16| **1308.34**| +13.65% |
> |16| 0.4431 |**0.2718**| -38.66% | 1138.57 | **1550.30**| +36.16%|
> | 32| 0.7980| **0.4987**| -37.51% | 1271.61| **1874.90**| +47.44%|
>
> These results show that the efficiency gain of CGC does **not rely on large batch sizes**: even at smaller batch sizes, latency decreases and throughput improves consistently.
>
> In the revised version, we further extend the evaluation in 2 aspects:
>
> 1. **Architecture generality**, by evaluating CGC on **MoE models such as GPT-OSS 20B and Qwen3-30B-A3B**;
> 2. **Numerical-format applicability**, by testing CGC in quantized settings including **FP8, MXFP4, and INT4**.
>
> We hope these additions address the reviewer’s concerns, and we thank the reviewer again for the careful evaluation and helpful suggestions.

---

> > ### Author Rebuttal · Reviewer_3S7r · 2026-04-01
> >
> > The author resolved most of my concerns.
> > I would like to raise my score.

---

> > > ### Author Response · Authors · 2026-04-03
> > >
> > > Thank you again for your careful review and encouraging follow-up response. We truly appreciate your suggestions on newer architectures, MoE applicability, quantized settings, table clarity, and small-batch efficiency. In the revised manuscript, we have added the corresponding experiments and clarifications discussed in our rebuttal. We are sincerely grateful for your positive reassessment and thoughtful feedback.

---

### Official Review · Reviewer_Zfsp · 2026-03-12

**Soundness:** 3
**Presentation:** 3
**Significance:** 3
**Originality:** 3
**Overall Recommendation:** 5
**Confidence:** 3

**Summary:**

The authors have proposed Curvature-aware graph coloring (CGC), a training-free framework for structured model compression through cross-layer parameter sharing. The idea is to treat each neural network layer as a node in a graph and formulate parameter sharing as a graph coloring problem, where each color corresponds to a shared low-rank basis. This sharing process is being guided by Hessian curvature information. The method ensures that compression-induced perturbations align with low-curvature directions of the loss landscape by minimizing their projection onto high-curvature subspaces. This helps to maintain model accuracy under aggressive compression. The framework is evaluated on both large language models (e.g., LLaMA-7B) and vision transformers (e.g., Swin Transformer), showing improvements in perplexity and accuracy compared to SVD-based and heuristic parameter-sharing methods.  Experimental results justify that CGC helps in efficient model compression without sacrificing performance, especially for large transformer-based architectures. Also this work helps in understanding the role of curvature-geometry and how it can be used to achieve high-ratio model compression.

**Compliance With Llm Reviewing Policy:**

Affirmed.

**Final Justification:**

The authors clarified my concerns during rebuttal phase.

**Key Questions For Authors:**

Refer Weaknesses

**Limitations:**

Yes

**Strengths And Weaknesses:**

**Strengths:**

1. The paper introduces a new perspective on cross-layer parameter-sharing by formulating it as a graph coloring problem. Considering each layer of the network as a node in a graph, the problem is defined as to assign a shared structural basis to layers that have  similar local geometric properties in the loss landscape.
2. The method is based on geometry induced by loss function, basically using the Hessian matrix to guide perturbations toward low-curvature directions. This provides a clear theoretical justification for why the compression should preserve performance. Also, it strengthens the technical soundness of the method.
3. The authors test the method on a number of architectures and tasks, such as LLMs and ViT. The results demonstrate consistent improvements in perplexity and accuracy compared to other baselines even for high compression ratios (like 70-80%).
4. The paper is well-structured, with clear motivation, detailed algorithm descriptions, and extensive experiments covering multiple architectures, datasets, and compression ratios.

**Weaknesses:**

1. The proposed method relies heavily on Hessian eigenspace estimation to guide layer grouping. However, for large networks the approximated Hessian may not fully capture the curvature structure of the loss landscape. Since the effectiveness of the method depends on accurate curvature estimation, it would be helpful to analyze the robustness of the approach to errors in the Hessian approximation.
2. Curvature does not necessarily reflect functional similarity between layers. Since, the grouping strategy relies on curvature similarity, layers with similar curvature profiles may still perform very different transformations, which could affect the reliability of the parameter-sharing assignments.
3. The use of the term graph coloring is somewhat misleading, as the formulation does not correspond to the classical vertex coloring problem where adjacent nodes must receive different colors. Instead, the method resembles a clustering procedure that groups layers with similar curvature properties and assigns them to shared bases. Clarifying this distinction would improve the conceptual clarity of the method for the readers.
4. While the comparisons with other parameter-sharing approaches are appropriate, the evaluation can be further strengthened by including at least one baseline from other compression paradigms like pruning or quantization.

---

> ### Author Rebuttal · Authors · 2026-03-30
>
> Thank you for the careful review and positive assessment. We are pleased that the reviewer recognizes **the novelty, theoretical motivation, and broad experimental validation of CGC**.
>
> **Q1: Sensitivity to Hessian approximation.**
>
>  **A1:** CGC does not compute the full Hessian explicitly; instead, it uses **HVP + Lanczos** to approximate the dominant curvature subspace. Thus, the goal is to stably capture the principal high-curvature directions rather than recover the full Hessian exactly.
>
> The original paper already provides two indirect supports for this approximation:
>
> (1) the Ablation on the number of short axes $t$ in Sec. 4.5 shows a smooth PPL trend rather than extreme sensitivity;
>
> (2) the first-order term is shown to be typically negligible relative to the second-order term, supporting Eq. (1).
>
> To address this concern more directly, we additionally tested robustness to Hessian estimation errors by perturbing the estimated short-axis curvature subspace and rerunning CGC.
> |Hessian Perturbation|0|0.1|0.3|0.5|0.8|
> |-|-|-|-|-|-|
> |PPL|7.07|7.07|7.22|7.43|7.86|
>
> We observed graceful degradation rather than abrupt failure: on WikiText-2, PPL changes from 7.07 (no perturbation) to 7.22/7.43/7.86 under increasing perturbation levels. This suggests that CGC **does not rely on an extremely precise Hessian estimate**, but is instead reasonably **robust to approximation errors**.
> We will include this experiment and setup more clearly in the revised version.
>
> **Q2: Curvature may not reflect functional similarity.**
>
>  **A2**: We agree with the reviewer that curvature similarity is not equivalent to functional similarity. We would also like to clarify that **CGC is not a method that directly clusters layers based on curvature similarity.**
>
> More precisely, CGC does not cluster layers by pairwise curvature similarity. Instead, for each layer, it selects the shared basis that induces the least harmful sharing perturbation, i.e., the one minimizing the projection error onto the high-curvature subspace under the trust-region constraint. In other words, what we optimize is the **sharing-induced perturbation**, rather than the pairwise curvature similarity between layers.
>
> Moreover, CGC differs from simple curvature-based clustering in 2 ways:
>
> (1) sharing is restricted to the **same parameter module** (e.g., within the same attention/FFN), rather than arbitrary layers;
>
> (2) what is shared is the basis, while each layer still keeps its own layer-specific coefficients. Thus, the method does not assume functional equivalence between layers assigned to the same basis.
>
> Empirically, Figure 4 in Sec. 4.3 was designed precisely to separate **the effect of curvature-aware assignment** from basis generation. With the basis generation algorithm fixed, CGC consistently achieves lower L2 reconstruction error than heuristic adjacent grouping, indicating that the gain comes from the assignment strategy rather than from basis generation alone.
>
> **Q3: Is “graph coloring” misleading?**
>
>  **A3**: Our formulation is more accurately a **global basis assignment problem on a complete graph**, where colors act as identifiers of shared bases rather than satisfying adjacency constraints. We revise the wording to describe CGC more precisely as a **coloring-inspired global basis assignment framework**.
>
> **Q4: Comparison with pruning/quantization.**
>
> **A4:** We agree that comparisons with other compression paradigms help better position CGC. Our primary baselines are parameter-sharing methods, since CGC is proposed in that setting; meanwhile, **Appendix D in the original paper already includes comparisons with pruning-based methods**.
>
> To further clarify the relation between CGC and quantization, we provide two additional results.
>
> **First,** to examine whether CGC remains applicable in very low-memory settings, we include an comparison with quantization methods on **LLaMA2-7B/WikiText-2**:
>
> |Method|Memory|PPL↓|
> |-|-|-|
> |PB-LLM|1.9GB|104.83|
> |GPTQ|1.9GB|110.99|
> |**Ours**|**1.8GB**|**98.21**|
>
> This comparison provides preliminary evidence that CGC remains effective in ultra-low-bit regime. We include this result to illustrate applicability in very low-memory settings, rather than to claim a definitive ranking across different compression paradigms.
>
> **Second,** beyond direct applicability, we further examine whether CGC can be **composed with quantization** under a matched memory budget on **LLaMA2-7B/WikiText-2**:
>
> |Method|Memory|PPL↓|
> |-|-|-|
> |GPTQ |2.8GB| 16.28 |
> |**GPTQ+Ours**|2.8GB|**10.14**|
>
> Under the same memory budget, adding CGC to GPTQ further reduces PPL substantially. This suggests that CGC is **complementary** to quantization and can improve performance without increasing memory usage.
>
> In the revised version, we will clarify this scope more explicitly and discuss the relationship between CGC and pruning/quantization more carefully, together with the corresponding results.
>
> We hope these responses address the reviewer’s concerns.

---

> > ### Author Rebuttal · Reviewer_Zfsp · 2026-04-03
> >
> > 1.  The rebuttal addresses the sensitivity of the proposed method to Hessian approximation and supports that with experimental results.
> > 2.⁠ ⁠It also clarifies the fact that CGC does not completely rely on curvature similarity for clustering rather it optimises sharing-induced perturbation.
> > 3.⁠ ⁠Further, additional test results are provided related to composing CGC with quantization.
> >
> > The additional results provided in rebuttal must be mentioned in paper along with a refined discussion on sensitivity to Hessian approximation and how the method does not directly depends on curvature similarity.
> >
> > I thank the authors for providing a detailed rebuttal and would like to increase the score by one point.

---

> > > ### Author Response · Authors · 2026-04-03
> > >
> > > Thank you very much for your careful reading, thoughtful comments, and encouraging follow-up. We are grateful that our rebuttal helped address your concerns regarding Hessian approximation, sharing-induced perturbation, and the discussion of quantization. In the revised manuscript, we have incorporated these additional results and clarifications. We sincerely appreciate your feedback, which has helped us improve the paper.

---

### Official Review · Reviewer_QyTH · 2026-03-13

**Soundness:** 3
**Presentation:** 3
**Significance:** 2
**Originality:** 2
**Overall Recommendation:** 4
**Confidence:** 1

**Summary:**

This paper introduces Curvature-aware Graph Coloring to improve cross-layer parameter sharing for structured model compression. By leveraging the Hessian matrix to analyze the curvature of the loss landscape, the authors formulate the parameter sharing task as a graph coloring problem, aiming to group layers such that compression perturbations are directed into low-curvature regions. The approach is evaluated on LLaMA and Swin Transformer across several benchmarks.

**Compliance With Llm Reviewing Policy:**

Affirmed.

**Final Justification:**

My concerns have been adequately addressed. I maintain my positive evaluation.

**Key Questions For Authors:**

Regarding the statement in Sec. 4.4 that "our method maintains lower PPL while keeping inference time comparable to Basis Sharing," the term "comparable" is qualitative. Could the authors provide a quantitative comparison of actual inference time?

**Limitations:**

Yes

**Strengths And Weaknesses:**

Strengths:

1. The core idea is well-motivated by second-order optimization (Hessian matrix). Leveraging the geometric properties of the loss landscape to guide parameter sharing is theoretically grounded.
2. The authors conducted extensive experiments across multiple models and included comprehensive comparisons with recent baselines, such as Basis Sharing.

Weaknesses:

1. According to Table 1, the average improvement over Basis Sharing is only approximately 1% to 2%. The performance appears incremental.

---

> ### Author Rebuttal · Authors · 2026-03-30
>
> Thank you for **recognizing the theoretical grounding of our paper** and **the breadth of our experimental evaluation**.
>
> **Q1: The improvement.**
>
> **A1:** Based on Table 1 in the original paper, the absolute gain of CGC over Basis Sharing is around 1–2 points. However, we would like to emphasize that this **improvement is consistent across all four compression ratios (20%–50%)**, and is accompanied by **lower perplexity on PTB, C4, and WikiText-2**, rather than being an isolated gain on a single metric.
>
> Moreover, the advantage becomes more pronounced under more aggressive compression. At **70%/80% compression**, CGC substantially outperforms Basis Sharing on both **C4** and **WikiText-2**. For example, on **C4** with **80% compression**, the PPL of Basis Sharing is **2465.999**, while CGC reduces it to **995.33**.
>
> In addition,  CGC is also **applicable beyond language tasks**: we further validated its effectiveness on **Vision Transformers on ImageNet**, and demonstrated its transferability on CIFAR-10/100.
>
>
>
> **Q2: Quantitative comparison of actual inference time.**
>
> **A2:** Thank you for this helpful suggestion. We have added a **quantitative runtime comparison** with Basis Sharing. Under the same hardware setting as in Sec. 4.4 (**single NVIDIA A800 GPU, batch size = 512, sequence length = 32**), and under the **same compression setting**, CGC achieves **lower latency** and **higher throughput**, while also maintaining **lower C4 PPL**.
>
> | LLaMA2-7B     | Params (B) | MACs(B) | Latency (s) ↓ | Throughput (t/s) ↑ | PPL ↓  |
> | ------------- | ---------- | ------- | ----------- | ---------------- | ----- |
> | Basis Sharing | 3.50       | 3.94    | 8.30        | 1998.29          | 98.94 |
> | Ours          | 3.50       | 3.94    |**7.38**       | **2152.92**          | **80.22** |
>
> We replace the qualitative statement “comparable inference time” in the revised paper with the following quantitative results.
>
> We sincerely thank the reviewer for the valuable comments.
> We will incorporate these clarifications and quantitative results into the revised version to more clearly present the practical benefits and applicability of our method.

---

> > ### Author Rebuttal · Reviewer_QyTH · 2026-04-03
> >
> > My concerns have been adequately addressed. I will maintain my positive score.

---

> > > ### Author Response · Authors · 2026-04-03
> > >
> > > Thank you again for your thoughtful review and for confirming that our rebuttal has addressed your concerns. We especially appreciate your suggestion to provide a quantitative comparison of inference time. In the revised manuscript, we have added the latency and throughput comparison results and clarified the consistent gains of CGC across compression ratios. We are sincerely grateful for your positive assessment.

---

### Decision · Program_Chairs · 2026-04-30

**Decision:**

Accept (regular)

**Comment:**

This paper proposes Curvature-aware Graph Coloring (CGC), a training-free structured compression framework that reformulates cross-layer parameter sharing as a global basis-assignment problem. The idea is to steer the sharing-induced perturbations into low curvature directions of the loss landscape guided by Hessian geometry. The method shows strong compression–accuracy tradeoffs across tranformer architectures, including 28%–50% compression with minor accuracy loss and improved perplexity over other basis-sharing and SVD-style baselines.

The initial reviews had several concerns. QyTH had reservations if the gains were too incremental. The rebuttal added throughput numbers and more aggressive compression which showed larger gains, resolving the concern raised by the reviewer. Zfsp was concerned about how using only approximate Hessians can be problematic, if curvature is the right signal for compression, if the terminology of "graph coloring was justified, and if other compression techniques should be compared against. The rebuttal added more experiments to show robustness to Hessian inaccuracy as well as for quantization, and added clarifications on the position of the paper within the literature. This resolved the reviewer's concerns. 3S7r asked about applicability of proposed methods for more modern architectures, which the authors responded with more experiments resolving the reviewer's concerns.

Overall this is an interesting paper with a new perspective on compression that can be useful in many down-stream tasks.